# Patient Satisfaction with Telemedicine during the COVID-19 Pandemic—A Systematic Review

**DOI:** 10.3390/ijerph19106113

**Published:** 2022-05-17

**Authors:** Karolina Pogorzelska, Slawomir Chlabicz

**Affiliations:** Department of Family Medicine, Medical University of Bialystok, 15-054 Bialystok, Poland

**Keywords:** telemedicine, patient satisfaction, telehealth

## Abstract

Telemedicine is a convenient tool for providing medical care remotely. It is routinely offered as an alternative to face-to-face consultations in healthcare settings all over the world. Due to the COVID-19 pandemic and increased use of telemedicine in everyday clinical practice, the effectiveness of this modality and patient satisfaction with telemedicine is a subject of growing concern. PubMed and Google Scholar databases were searched. Papers published between January 2020 and August 2021 which met inclusion and exclusion criteria were analyzed. During the COVID-19 pandemic patients have found telemedicine a beneficial tool for consulting healthcare providers. A high level of satisfaction with telehealth was observed in each study across every medical specialty. Telemedicine is undoubtedly a convenient tool that has helped ensure continuity of medical care during the COVID-19 pandemic thanks to its considerable potential. In particular situations, telehealth may adequately replace face-to-face consultation. Regular patients’ feedback is necessary to improve the use of telemedicine in the future.

## 1. Introduction

Telemedicine refers to the practice of providing medical care remotely. It is routinely offered as an alternative to face-to-face consultations in healthcare settings all over the world [1,2,3,4]. A number of investigations have revealed that remote consultations improve access to healthcare and reduce the workload of healthcare workers [1,2,3]. Furthermore, better prevention of acute infectious diseases is observed with the use of telehealth, which has been of particular importance during the COVID-19 pandemic. In August 2021, the total number of COVID-19 cases exceed 200 million worldwide, just six months after reaching 100 million [5]. The outbreak of the COVID-19 pandemic has had a profound impact on healthcare delivery globally. A number of countries have faced challenges related to healthcare provision, including staff shortages, poor distribution of services, and incompatibility between the health needs of the population and the competencies of health professionals [6]. Many countries have reported a more or less severe disruption in the treatment of hypertension, diabetes and diabetes-related complications, and cardiovascular emergencies [7]. Countries that have reported disruptions in the availability of medical services, have implemented alternative strategies such as telehealth to ensure continuity of medical care. Due to the COVID-19 pandemic and increased use of telemedicine in everyday clinical practice, the effectiveness of this modality and patient satisfaction with telemedicine is a subject of growing concern. In our systematic review, we focused on the patient perspective and the level of satisfaction among patients suffering from different medical conditions with telemedicine during the COVID-19 pandemic.

## 2. Materials and Methods

In August 2021, we searched PubMed and Google Scholar online databases for the following search terms: “telemedicine”, “teleconsultation”, “telehealth” in combination with “patient’s satisfaction”, “patient’s perspective”, “patient’s experience”. We did not search MEDLINE as we wanted to avoid duplicating papers already identified in PubMed and Google Scholar. We found a total of 437 articles in PubMed and Google Scholar. Then, we consecutively screened abstracts and full-text articles which assessed patient satisfaction during the COVID-19 pandemic. We excluded articles which did not meet inclusion and exclusion criteria. The inclusion criteria were as follows: survey studies including adults, original studies published between January 2020 and August 2021, studies published in English. The exclusion criteria were as follows: studies published before January 2020, systematic reviews and meta-analyses, studies that did not evaluate patient satisfaction (Figure 1). Finally, 53 papers were analysed in this review.

Identified studies covered mainly the patient’s satisfaction with telemedicine during COVID-19 pandemic across an extensive range of medical specialties: Neurosurgery [8,9,10,11], Primary Care [12,13], Dermatology [14], Obstetrics [15], Orthopedics [16,17,18], Oncology [19,20,21,22,23,24,25], Rheumatology [26,27,28,29], Otolaryngology [30,31,32,33,34,35], Urogynecology [36], Cardiology [37], Psychiatry [38], Internal Medicine [39], Allergology [40,41], Endocrinology [42], Gastroenterology [43], Spinal Disorders [44,45,46], Oral and Maxillofacial Surgery [47], Surgery [48,49,50,51], Ophthalmology [52,53], Urology [54,55], Emergency Room [56]. Additionally, eight studies assessed caregivers’ satisfaction and experiences with telehealth. Three studies evaluated satisfaction with the use of telemedicine among new patients compared to follow-up patients. Six of the cited papers assessed whether the level of patient satisfaction differed between a virtual consultation and a traditional face-to-face encounter. Three studies aimed to establish the rate of missed appointments for the purpose of facilitating the future use of teleservice in medicine. The reviewed studies were conducted in the United States (34), United Kingdom (8), Australia (2), Spain (2), India (2), France (1), Italy (1), New Zealand (1), North Macedonia (1) and Republic of Korea (1).

## 3. Results

### 3.1. Satisfaction Rate

During the COVID-19 pandemic, patients have found telehealth a valuable tool for consulting healthcare providers. Table 1 summarizes studies which examined the level of satisfaction among patients suffering from different medical conditions. A study conducted by Tanya Ngo in a Student-Run Free Clinic revealed that most patients (97.6%) were satisfied with their telehealth experience [57]. Additionally, a survey of 1010 respondents revealed that overall satisfaction with telehealth in primary care was high—91% of respondents were satisfied with video consultations and 86% were happy with telephone consultations [13]. A study by Ashwin Ramaswamy et al., in which the surveyed group of patients was largest in comparison to patient groups examined in other papers analysed in the present review, patient satisfaction with video consultations was significantly higher in comparison to in-person visits (94.9% vs. 93.0%, *p* < 0.001) [58]. A different study demonstrated that trust in doctors correlated with higher patient satisfaction with remote visits [39]. 88% of patients agreed that a virtual consultation was more convenient for them than an in-person visit [8]. In a study by Park et al., 87.1% of patients thought telemedicine had the same reliability as in-person visits [56]. A study by Porche et al. reported interesting findings that indicate that virtual consultations are not significantly different from in-person visits in all domains (*p* = 0.085) [10].

### 3.2. Other Aspects of Virtual Consulations

#### 3.2.1. Missed Appointment Rate

In the era of the COVID-19 pandemic, it is important that the economic impact of no-show rates is evaluated. Three of the cited studies examined the in-office visit no-show rate compared to the telehealth visit no-show rate. In a study conducted by Brenden Drerup, the in-office missed appointment rate was 36.1% compared with the telehealth visit no-show rate of 7.5%. This was highly statistically significant at *p* < 0.0001 identified by Fisher’s exact test [12]. Analogous results were observed in research by Sumithra Jeganathan, which showed a significantly lower rate of appointments canceled by patients for in-person visits than telehealth visits (5.44% vs. 3.82%, *p* = 0.021). Telehealth visits also had a lower no-show rate; however, this difference was not statistically significant [15].

Therefore, the use of telehealth is not only beneficial for patients but also increases the efficiency of healthcare systems by reducing missed appointment rates.

#### 3.2.2. Clinical Needs of Patients

A large number of the reviewed studies revealed that telehealth visits adequately addressed patients’ needs [16,23,34,41] What is noteworthy, in two of the analysed studies, all respondents confirmed that their diagnosis and treatment options were satisfactorily explained by their doctors who spent sufficient time with them. [48,54] Regarding privacy concerns, there was general acceptance of virtual consultations as an alternative to face-to-face encounters.

#### 3.2.3. Technical Aspects

All reviewed papers reported a significant level of satisfaction with connection quality and ease of use of telemedicine [8,15,21] However, patients who encountered connection and other technical issues expressed lower satisfaction with virtual visits [19].

#### 3.2.4. Time and Cost Savings

Saving on travel represented the most important advantage of having a virtual consultation, followed by time savings and cost savings, and reduced family interruption. [24,26,42,54]. Many patients emphasized that they did not have to leave their workplace to consult with their doctor [43]. Furthermore, decreasing the exposure to SARS-CoV-2 infection is the perceived benefit of telemedicine [11].

#### 3.2.5. Willingness to Use Telehealth in the Future

Preference for future telehealth visits was reported by a significant number of the reviewed studies [8,13,16,21,22,39,42,52,54]. This suggests that respondents received appropriate care and were satisfied with the telemedicine encounter [21]. Patients who preferred a virtual consultation generally had a greater distance to travel, were younger and had fewer problems with their physical examination in comparison to patients who were more inclined to choose in-person visits [27,36,44]. It is worth emphasizing that many patients would like to be offered the choice between an in-person visit and a virtual consultation in the future [26].

The results mentioned above have been summarized in Table 2.

## 4. Discussion

### 4.1. Advantages of Telemedicine

Due to the COVID-19 pandemic, telemedicine has become a tool used to maintain continuity of medical care. The advantages of telehealth are the comfort and convenience it offers patients, and a reduction in costs incurred by the healthcare system. By triaging patients through a telehealth platform, access to primary and specialty care is increased [38], and medical providers’ workload may be decreased. Most importantly, nonessential in-person appointments in the clinic or surgery may be replaced with a virtual consultation to reduce patients’ exposure to acute infectious diseases such as COVID-19. Moreover, human and equipment resources could be redirected to fight against the COVID-19 pandemic [8]. The articles reviewed in the present paper reported a high level of patient satisfaction with telemedicine encounters. Furthermore, the studies revealed that new patients reported higher satisfaction scores than follow-up patients [14]. Similarly, established patients were less satisfied than new patients. This may be related to previous experience since established patients may consider in-person visits to be more reliable than virtual consultations. In a study by Firas Hentati, 62.2% of respondents indicated that they did not prefer their telemedicine encounter to an in-person visit [32]. On the other hand, in a study by Janet S. Choi, follow-up visits and postoperative visits were associated with higher patient satisfaction levels in comparison to new patient encounters [33]. The results of a study by Sheena Bhuva demonstrate that telehealth can be a tool to provide satisfactory and effective care, particularly with follow-up visits, in the case of which a greater number of patients preferred a virtual consultation to a face-to-face encounter [45]. Some patients found virtual visits more successful when there was a pre-existing relationship with the doctor [13,18]. Moreover, the majority of respondents believe that a virtual consultation could adequately replace an in-person visit [38]. It should be mentioned that the rate of missed virtual visits is significantly lower compared with in-person encounters, which has financial implications [12]. The time and cost savings of having a consultation with a medical professional from one’s workplace or home is one of the other advantages of telemedicine emphasized by respondents [20].

### 4.2. Limitations of Telemedicine

It needs to be remembered that telemedicine has its limitations. One of the biggest concerns faced by patients is the lack of a physical examination, which may lead to misdiagnosis [13,17]. Moreover, the absence of direct physical examination reduces the patient’s preference for virtual consultations in the future [44]. Hence, telemedicine is more often recommended for follow-up visits. In one out of the reviewed papers, patients reported that they had not been sufficiently asked about their medical history or they had not spent enough time with their doctors. Establishing rapport between the doctor and the patient is another important factor in favor of an in-person encounter. Although few patients experienced difficulty connecting with their doctor, the use of telemedicine may be challenging for those not well acquainted with new information and communications technologies or those without access to appropriate equipment, such as elderly patients. Patients who had video consultations instead of telephone consultations were statistically significantly more likely to be under the age of 65 (*p* = 0.0031) [12]. Respondents who experienced smartphone data or internet connection problems reported lower levels of satisfaction [19].

Security and protection of medical data, and patient privacy are also among the main concerns related to telemedicine [59]. There is no doubt that a secure computer system allows access to medical data by authorized personnel and prevents any unauthorized access. Another aspect that needs to be carefully assessed is obtaining verbal informed consent which should be noted in the patient’s medical records [60]. Effective verification and confirmation of the patient’s identity is an essential part of ensuring patient privacy and preventing imposture. Due to the occurrence of cyberattacks in the past, doctors should inform patients about potential risks associated with telehealth services. Telehealth platforms have healthcare-specific features and security. However, it is sometimes necessary to use video conferencing software, e.g., Skype for Business, Microsoft Teams, VSee, Doxy.me. which is considered highly secure by the U.S Department of Health and Human Service (HHS) [61]. By way of illustration, in a study by Mojdehbakh, the surveyed patients reported that their privacy was respected as either “excellent” (84.0%, *n* = 95) or “good” (8.8%, *n* = 10) during virtual consultations [21].

The findings of the present review are limited by the nature of the examined studies—most of the research was conducted in a single clinic, where patients were treated by a few doctors. Due to the low survey response rate in the reviewed studies, the results do not correspond to the entire population of patients participating in telemedicine visits. The pandemic and related stressors may have also influenced the response rate. Furthermore, the fact that the reviewed studies were conducted during the COVID-19 pandemic, when patients may have been reluctant to attend in-person appointments, could have led to a higher patient satisfaction rate with telemedicine. Therefore, it cannot be presumed that the reported data represent patient attitudes to telemedicine beyond the pandemic. The majority of respondents in the reviewed studies were women, who are more likely to complete surveys than men. Women also visit primary healthcare centres more frequently than men. Some studies revealed that the satisfaction rate was higher among women in comparison to men, which may have impacted the results obtained in those studies. Patients who were more acquainted with new information and communication technologies were more likely to participate in the surveys, which improved satisfaction scores.

To sum up, during the pandemic telehealth offered support to traditional medicine with high patient satisfaction. The utilization of telemedicine services allowed us to prevent the spread of the SARS-CoV-2 virus and played an important role in maintaining the continuity of healthcare.

## 5. Conclusions

Telemedicine is undoubtedly a convenient tool that has helped maintain continuity of medical care during the COVID-19 pandemic thanks to its considerable potential. In particular situations, telehealth may adequately replace face-to-face consultation. Regular patient feedback is necessary to improve the use of telemedicine in the future.

## Figures and Tables

**Figure 1 ijerph-19-06113-f001:**
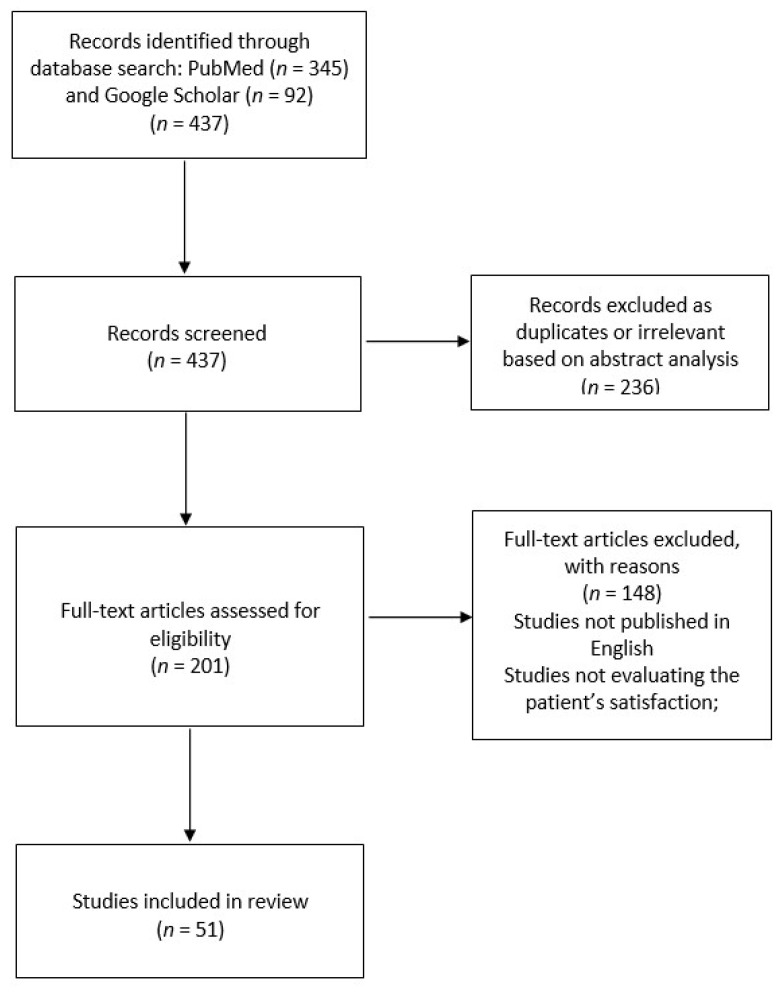
Search and study selection process.

**Table 1 ijerph-19-06113-t001:** Summary of the studies on patient’s satisfaction with telemedicine.

Author	Sample	Clinical Description	Country	Outcome Measure	Patient Satisfaction
Bryan A. Johnson et al.	*n* = 785 patients Response rate = 38.4%	Oncology Clinic	USA	40—questions survey with a 7-point Likert scale ranging responses	The median patient satisfaction score was 5.5 (interquartile range [IQR] 4.25–6.25). The median telemedicine usability score was 5.6 (IQR 4.4–6.2). A strong positive correlation was seen between satisfaction and usability, with a Spearman correlation coefficient (*ρ*) of 0.80 (*p* < 0.001) [19]
Mohamed E. Ahmed et al.	*n* = 52 patients	Advanced Prostate Cancer Clinic	USA	28—item survey; patient satisfaction were measured with both closed 5—point Likert scale ranging responses,	The median degree of satisfaction was 9 on a 10-point scale, with 10 being the highest level of satisfaction [54]
Elisa Picardo et al.	*n* = 345 patients	Oncology Clinic	Italy	22—item survey with both closed 6—point Likert scale ranging responses	The results of the present study show that patients with pelvic cancer were more satisfied with telemedicine compared to patients in the breast cancer group (*p* = 0.058). Moreover, the breast cancer group reported telemedicine as comparable as face-to-face consultation more than pelvic cancer group, even though they were overall less satisfied (*p* = 0.0001) [20]
Rachel P. Mojdehbakhsh et al.	*n* = 113 patients Response rate = 74.8%	Gynaecologic Oncology Clinic	USA	10—question survey with patient satisfaction rate measured using 4—point scale	The overall telemedicine treatment experience was rated as excellent by 63.7% of patients (*n* = 72), good by an additional 24.8% (*n* = 28). Overall, 82.3% of patients endorsed using telemedicine again and 85.8% would recommend using telemedicine to another person [21]
Ajithray Sathiyaraj et al.	*n* = 70 patients Response rate = 21%	Cancer Center	USA	11—questions survey	73% of cancer patients undergoing prechemotherapy evaluation were satisfied with the video visit experience. 70% of patients believed that video visits were as good as in-person visits, but 65% of patients stated that an in-person visit was their preferred method [22]
Philip J. Chang et al.	*n* = 76 patients Response rate = 45.8%	Rehabiliation Cancer Center	USA	7—items survey with 5—point Likert scale ranging	94.8% of patient responses reported “quite a bit” or “very much” for the telemedicine visit being a good experience; 63.1% of patient responses reported “quite a bit” or “very much” for interest in using telemedicine visits in the future [23]
Michael Meno et al.	*n* = 212 patients Response rate = 49.8%	Cancer Clinic	USA	5—item survey with 5 -point Likert scale ranging responses;	Most patients felt that the overall quality of the telehealth visit was the same as that experienced with an office visit (55.2%) or better (10.4%). Half of the patients (50%) found the personal connection in office visits to be better [25]
Adeel Abbas Dhahri et al.	*n* = 43 patients	Surgery Clinics	United Kingdom	12—question survey with 5—point Likert scale measuring patient satisfaction	Overall experience with most of the patients was positive (41/43; 95.34%). All (100%) patients thought that the video telemedicine solution met their needs. 93% of respondents recommended to use it for future consultations [48]
T. J. Horgan et al.	*n* = 109 patients Response rate = 80.74%	Oral and Miaxillofacial Surgery Department	USA	16—question survey (G-MISS) with 5—point Liker scale ranging and additional survey	The total mean (SD) G-MISS score for satisfaction was high at 82.12 (7.96) indicating a high level of satisfaction among all patients. 83.48% found telephone consultation to be as convenient [47]
Alicia Ruiz de la Hermosa et al.	*n* = 1706 patients	General Surgery Clinic	Spain	5—question survey; patient satisfaction measured with 10—point scale ranging responses	The overall satisfaction was 8.7 out of 10. 37.2% would preferred a face-to-face visit because of difficulties with the teleconsultation [49]
Aminah Sallam et al.	*n* = 50 patients Response rate = 57.5%	Cardiac Surgery Clinic	USA	24—item survey; patient satisfaction measured with 5—point Likert scale ranging;	Patients described the ability of their surgeons to diagnose problems and the thoroughness and skill of their surgeon in treating their conditions as good to excellent (mean score of 4.3, SD 0.9). Patients expressed satisfaction with the care they received (mean score 4.8, SD 0.5) [50]
Elvina Wiadji et al.	*n* = 1166 patients Response rate = 12.3%	Surgery Clinic	Australia	29—item survey; patient satisfaction measured using 4—point scale ranging responses;	The majority of patients (94%), were satisfied with the quality of their surgical telehealth consultation and 75% felt it delivered the same level of care as face-to-face encounters [51]
Brenden Drerup et al.	*n* = 65 patients Response rate = 67.7%	Primary and Specialty Care Clinic	USA	9—question survey using both closed 5—point Likert scale responses	Patients who experienced telehealth visits had similar responses to seven of nine satisfaction metrics when compared with those who attended in-office visits [12]
Fiona Imlach et al.	*n* = 1010 patients Response rate = 84.87%	Primary Care Clinic	New Zealand	Online patient survey with open-ended questions	Overall satisfaction with telehealth was high, at 91% for video and 86% for telephone consultations, but was slightly lower than in-person visits (92%) [13]
Ashwin Ramaswamy et al.	*n* = 38.609 patients	Outpatient Clinic	USA	19—item survey with 5—point Likert scale ranging responses	Patient satisfaction with video was significantly higher than in-person visits (94.9% vs. 92.5%, *p* < 0.001) [58].
Tanya Ngo	*n* = 86 patients	Student—Run Free Clinic	USA	Survey with patient satisfaction rate measuring with 8—question with 3—point Likert scale ranging	Overall, most patients reported feeling satisfied with their telehealth experience (97.6%). However, only 64 patients (74.4%) felt that their telemedicine visits were as good as in-person clinic appointments [57]
Hyung Youl Park et al.	*n* = 906 patients Response rate = 13.2%	Outpatient Clinic and Emergency Room	Republic of Korea	5—item patient survey;	Overall satisfaction was reported by 86% of patients, whereas only 52.7% of doctors were satisfied with telemedicine (*p* = 0.000 for both doctors and nurses compared with patients). 87.1% of patients thought telemedicine had the same reliability as in-person visits [56]
Lucinda Adams et al.	*n* = 128 patients Response rate = 29%	Rheumatology Clinic	Australia	18—questions survey using both closed 5-point Likert scales ranging responses and four free-text questions	61.7% agreed or strongly agreed when asked, ‘In general, I am satisfied with the telemedicine system’ [26].
Mahta Mortezavi et al.	*n* = 359 patients Response rate = 70.1%	Rheumatology Clinic	USA	3—item survey with both closed 5—point Likert scale ranging responses	The majority of patients (74%) were satisfied with their virtual visit, but they were more likely to be satisfied if their visit was over video rather than phone. They preferred an in-person visit if they were meeting a doctor for the first time [27]
Jesus Tornero—Molina et al.	*n* = 469 patients	Rheumatology Clinic	Spain	Survey with 10—point scale measuring patient satisfaction	The mean levels of satisfaction with the rheumatology teleconsultation procedure were very high (8.62). Over 80% of patients attended would repeat the teleconsultation and 79.3% considered them useful. The need for a face-to-face consultation after teleconsultation was not analyzed [28]
Matthew T. Jones et al.	*n* = 297 patients	Rheumatology Clinic	United Kingdom	30-item questionnaire consisted of single and Likert scale responses.	Overall, 150 (52%) and 69 (24%) responses indicated satisfaction with the telephone consultation by either agreeing or strongly agreeing, respectively. 60% would be happy to have future routine follow-up telephone consultations [29]
Eleanor Layfield et al.	*n* = 100 patients Response rate 68.49%	Head and Neck Otolaryngology Clinic	USA	21—questions survey with both closed 7—point Likert scale ranging responses	The total average score was 6.01. The highest scores were for questions related to satisfaction with telehealth (6.29), while the lowest was related to reliability (4.86) [30]
Janet S. Choi et al.	*n* = 407 patients Response rate = 19%	Otolaryngology Clinic	USA	14—item survey with 5—point Likert scale ranging (TQS)	Mean Press Ganey patient satisfaction scores during COVID-19 remained high at 94.5 (SD, 8.8; range, 20–100) for telemedicine visits. Patient satisfaction (TSQ score) for all telemedicine encounters was high at 4.17 (SD, 0.2; range, 1–5). Mean (SD) TSQ scores were 4.2 (0.67) and 3.67 (0.83) for videoconference and telephone encounters, respectively [33]
Phoebe Elizabeth Riley et al.	*n* = 325 patients	Otolaryngology Clinic	USA	10—question survey with 5—point Likert scale ranging	Patients demonstrated the high satisfaction with telemedicine (average score, 4.49 of 5). High satisfaction was consistent across groups for distance to travel, age, and reason for referral [34].
M. Fieux et al.	*n* = 100 patients	Otolaryngology Clinic	France	12—question survey with 5—point Likert scale	98% of patients responded that the physician had answered all of their questions, while 49% felt teleconsultation was not equivalent to face-to-face consultation [35]
Andrew M. Rizzi et al.	*n* = 299 patients Response rate = 66.4%	Orthopaedic Clinic	USA	standardized validated post-visit satisfaction survey	Not recorded [16]
Sandeep Kumar et al.	*n* = 450 patients Response rate = 88.67%	Orthopaedic Clinic	India	6—questions survey	The overall satisfaction-rate to telemedicine was 92%, and only 7.2% of patients had difficulty in understanding or following telemedicine-based advice [17]
Lesslie J. Bisson et al.	*n* = 2049 patients	Orthopaedic Practise	USA	9—question survey with 5—point Likert scale ranging. The Patient Satisfaction Aggregate (PSA) score was estimated by taking the average of the five questions stated above with 5-point interval scales and transformed to a 0 to 1 continuous scale	No significant differences between modes of visit were observed for explanation (*p* = 0.22), spending enough time (*p* = 0.23), overall service from physician (*p* = 0.28), recommend to others (*p* = 0.59), call center (*p* = 0.49), physical therapy (*p* = 0.75), physician staff (*p* = 0.16), or billing staff (*p* = 0.23) [18]
Megan V. Morisada et al.	*n* = 69 patients	Rhinology Clinic	USA	18—item survey with 5—point Likert scale ranging responses	There was no difference in patient satisfaction between the virtual visits (mean total sum score = 78.1) and clinic visits (mean total sum score = 78.4) groups (*p* = 0.67) [31]
Firas Hentati et al.	*n* = 45 patients Response rate = 42.1%	Rhinology Clinic	USA	The survey consisted of 8 close—ended and 3 open—ended question on patient experience and satisfaction	80% of patients stated that their needs were met during their telemedicine visit. 77.8% of respondents declared they would do another virtual visit in the future if the pandemic ends [32]
S.M. Shahid et al.	*n* = 53 patients	Ophthalmology Clinic	United Kingdom	11—question survey with patient satisfaction rate measuring with 5—point Likert scale	The patients scored the overall satisfaction at a mean of 4.3 out of 5. 91% of respondents felt that they received the appropriate advice regarding the postoperative drop regime [52]
Vidushi Golash et al.	*n* = 120 patients	Oculoplastic Clinic	United Kingdom	20/22—question survey with 10—point scale measuring convenience and patient satisfaction	55 % of telephone and 82.5% of video consultation patients felt face-to-face reviews would not have changed the appointment outcome. Satisfaction scores of 10/10 were given by 71.3% of telephone and 72.5% of video consultation patients [53]
Alexander M. Satin et al.	*n* = 772 patients Response rate = 21.9%	Spine Surgeon Private Clinic	USA	8—questions survey with patient satisfaction rate measured using 5—point Likert scale	Overall, 87.7% of patients reported that they were satisfied with their telemedicine visit with 70% reporting a score of 5 out of 5 (“very satisfied”) [44]
Sheena Bhuva et al.	*n* = 172 patients Response rate = 25%	Spine Physical Medicine and Rehabilitation Clinic	USA	10—questions survey with patient satisfaction rate measured using 5—point Likert scale	97.6% were very satisfied or satisfied (83.7% of the patients were very satisfied) with their telemedicine appointment. 64.5% of the patients preferred telemedicine over in-person appointments [45]
Karim Shafi et al.	*n* = 84 patients Response rate = 76%	Spinal Disorders Facility	USA	7—question survey with 5—point Likert scale measuring patient satisfaction	81.0% of respondents reported that they were “extremely satisfied” (5/5) with their visit. 75.6% of patients noted that they were “extremely satisfied” with their treatment plan, with a mean score of 4.71 (SD = 0.55) [46]
Amol Raheja et al.	*n* = 231 patients	Neurosurgery Clinic	India	16—question survey with patient satisfaction measured using 5—point Liker scale ranging responses	The majority of the respondents (58%) either agreed/strongly agreed that teleconsultation helped them tide over the medical exigency during the lockdown; however, the clinical diagnosis did not influence this response (*p* = 0.21). The vast majority of the respondents felt that teleconsultation is beneficial (97%), as it minimizes their exposure to COVID-19 [11].
Ken Porche et al.	*n* = 698 patients	Neurosurgery Clinic	USA	9—question surveys with patient satisfaction rate measured using 5—point Likert scale	Patient overall satisfaction score was slightly higher with telemedicine visits compared to in-person corrected for care provider differences (94.2 ± 12.2 vs. 93.1 ± 13.4, *p* = 0.085) [10].
Alina Mohanty et al.	*n* = 122 patients Response rate = 20%	Neurosurgery Clinic	USA	13—question survey using both closed 5—point Likert scale responses and multiple choice responses	92 % of patients were satisfied with particular telehealth visit. 88 % of patients claimed that telehealth visit was more convenient for them than an in-person visit [8]
Elise J. Yoon et al.	*n* = 310 patients Response rate = 52.6%	Neurosurgery Division	USA	Survey using both closed 7—point Likert scale ranging responses	The mean overall satisfaction score was 6.32 ± 1.27. Subgroup analyses by new/established patient status and distance from their home to the clinic showed no significant difference in mean satisfaction scores between groups [9]
S. Shahzad Mustafa et al.	*n* = 177 patients Response rate = 61%	Allergology and Immunology Practise	USA	Survey using both closed 5—point Linkert scale ranging responses	Nearly 97% of patients were satisfied with their telemedicine encounter, and 77.4% believed it was as satisfactory as an in-person encounter [40]
Kasey Lanier et al.	*n* = 162 patients Response rate = 58%	Allergology Clinic	USA	6—question survey with responses scaled from 0 to 10 (the highest satisfaction score). The scale was dichotomized as a response of 10 versus less than 10, in accordance with other measures of patient satisfaction.	Overall, 88% of patients rated their comfort level seeing a doctor via telemedicine as a 10. 93% of respondents stated that their doctor explained their condition in an easily understood manner. 77% of patients would strongly recommend telemedicine services to others [41]
Judy Hamad et al.	*n* = 184 patients Response rate = 63.88%	Dermatology Clinic	USA	25—questions survey using both closed 12 point Likert scale responses	86.4% (159/184) of participants reporting positive overall satisfaction and experiences with teledermatology. New patients had significantly higher Likert scores for overall satisfaction with teledermatology than those of follow-up patients (new patients: mean 4.70; existing patients: mean 4.43; *p* = 0.03) [14]
Sumithra Jeganathan et al.	*n* = 91 patients Response rate = 10.6%	Obstetric Clinic	USA	11—questions survey using both closed 6 point Likert scale responses	Overall, 86.9% of patients were satisfied with the care they received and 78.3% would recommend telehealth visits to others [15]
S. Glass Clark et al.	*n* = 94 patients	Urogynecology Office	USA	6—questions survey using both closed 5—point Likert scales ranging responses and additional comments or concerns	The majority of patients answered either “agree” or “strongly agree” with the statement “All of my questions and concerns were addressed to my satisfaction during my video visit” (*n* = 89, 94.7%) [36].
Kadri Haxhihamza et al.	*n* = 28 patients	Psychiatry Clinic	North Macedonia	anonymous 18—questions survey using both closed 5—Likert scale responses	Overall satisfaction with psychiatric care was high (80.22%) [38]
Devinder Kaur et al.	*n* = 106 patient Response rate = 61.3%	Diabetes and Endocrinology Clinic	United Kingdom	Survey using the 5 -point Likert scale ranging responses	Overall, 97% of respondents indicated that they were satisfied with the quality of service being provided via telemedicine [42]
Zia Rahman et al.	*n* = 98 patients Response rate = 49.7%	Gastroenterology Clinic	United Kingdom	15—item survey; patient satisfaction were measured with 5—point Likert scale	High satisfaction scores were reported more in patients who had a prior consultation in a face-to-face clinic, i.e., follow-up group (51/66; 77.3%), than patients who had audio-consultation in their first clinic visit, i.e., new patient (15/32; 46.9%) [43]
Sharon Orrange et al.	*n* = 368 patients Response rate = 22.7%	Internal Medicine Clinic	USA	11—question surveys with patient satisfaction rate measured using 5—point Likert scale	Across the study, respondents were very satisfied (173/365, 47.4%) or satisfied (*n* = 129, 35.3%) with their telemedicine visit. Higher physician trust was associated with higher patient satisfaction (Spearman correlation r = 0.51, *p* < 0.001) [39]
Allanah Smrke et al.	*n* = 108 patients	Sarcoma Unit	United Kingdom	Patient experience survey (online or paper)	Mean satisfaction with telephone consultation was higher than face-to-face consultation (rating 8.99/10 v 8.35/10, respectively). The majority of patients (*n* = 86; 80%) indicated that they would like at least some future appointments to be performed using telemedicine [24]
Aniruddha Singh et al.	*n* = 120 patients Response rate = 11%	Cardiology Clinic	USA	24—item survey; patient satisfaction measured with 5—point Likert scale ranging responses	The no-show rate for telehealth visits (345/2019, 17%) was nearly identical to the typical no-show rate for in-person appointments. (17%). Both in-person and telehealth were viewed favorably, but in-person was rated higher across all domains of patient satisfaction [37]
Agathoklis Efthymiadis et al.	*n* = 119 patients	Urology Clinic	United Kingdom	7—question survey with 4—point Likert scale ranging responses;	The majority of responses to the adapted survey (Q1–7) were graded as ‘Excellent’, ranging from 79 (66%) to 112 (94%). ‘Agree’ responses ranged from 92 (77%) to 117 (98%) for questions (Q8–12), indicating high satisfaction [55]

**Table 2 ijerph-19-06113-t002:** Summary of the studies on other aspects of telemedicine.

Author	Patient Clinical Needs	Willingness to Use Telemedicine in the Future	Technical Aspects	Time Save	Missed Appointment Rate
Bryan A. Johnson et al. [19]	Not recorded	Not recorded	During the telemedicine visit, most of the patients did not report any connection trouble (61.3%)	Patient characteristics, including location of residence (*p* = 0.421) was not significantly related to satisfaction score.	Note recorded
Mohamed E. Ahmed et al. [54]	All patients verified that their physician had adequately explained their diagnosis and treatment options.	94% of the patients shared that they would participate in a future teleconsultation if it was offered	The most patients (94%) agreed that they were able to hear (and see) their physician clearly.	Patients reported that saving on travel represented the most important advantage of having virtual consultations,	Not recorded
Elisa Picardo et al. [20]	The breast cancer group reported telemedicine as comparable to a face-to-face appointment more than the pelvic cancer group, even though they were overall less satisfied (statistically significant, *p* = 0.001)	Not recorded	Not recorded	Not recorded	Not recorded
Rachel P. Mojdehbakhsh et al. [21]	A total of 68.1% (*n* = 77) of patients felt that the explanation of treatment by the telemedicine staff was excellent and an additional 25.7% (*n* = 29) said the explanation was good.	Overall, 82.3% of patients endorsed using telemedicine again.	There was overwhelming satisfaction with voice quality of the encounter as 75.2% (*n* = 85) responded excellent and 21.2% (*n* = 24) responded good.	Not recorded	Not recorded
Ajithray Sathiyaraj et al. [22]	While most patients (70%) reported that video visits were just as good as in-person visits, none said that they were better.	80% of patients also reported that they probably or definitely would use video visits if it were an option in the future.	Not recorded	Not recorded	Not recorded
Philip J. Chang et al. [23]	Satisfaction tended to be marginally higher when encounters were for stable or were conducted through video versus phone.	43.3% of patients are interested in using phone/video visits in the future	Not recorded	Not recorded	Not recorded
Michael Meno et al. [25]	Most patients felt that the overall quality of the telehealth visit was the same as that experienced with an office visit (55.2%) or better (10.4%)	Preference for wanting some future visits to be telehealth was seen in only 57.1% of patients.	Not recorded	Not recorded	Not recorded
Adeel Abbas Dhahri et al. [48]	All (100%) patients thought the video telemedicine solution met their needs.	Majority (*n* = 40; 93.02%) of the patients of the patients opted to choose video consultation for future again	Majority of the patients were pleased with the sound quality (33; 76.74%) and the video quality (34; 79.06%).	Not recorded	Not recorded
T. J. Horgan et al. [47]	Over ninety percent of patients felt as able to ask questions and 94.49% understood the information given just as easily.	Overall, 83.48% of patients said they would be willing to have a telephone consultation in future.	Not recorded	83.48% of patients found telephone consultation to be as convenient.	Not recorded
Alicia Ruiz de la Hermosa et al. [49]	The 73.6% considered that teleconsultation was able to fully or partially resolve the reason for their medical appointment	Not recorded	37.2% would preferred a face-to-face visit because of difficulties with the teleconsultation.	Not recorded	Not recorded
Aminah Sallam et al. [50]	74 % of Patients who received telemedicine consultation and 57% of patients who received video consultation, rated their surgeon’s ability to diagnose problems as excellent.	12% telemedicine and 39% video consultation patients said they would prefer their next postoperative appointment to be via telemedicine even if there was not a stay-at-home order in place.	The majority of respondents could hear their surgeon clearly.	Of the patients who said they preferred telemedicine visits, 79% reported an average travel distance of 23 miles, 96% reported an average travel time of 32.5min, and 87% reported an average cost of $6.70.	Not recorded
Elvina Wiadji et al. [51]	Most patients were satisfied with the quality of their telehealth consultation (94%)	Telephone consultations would be considered by 34% and video-link consultations by 49%	Only 63 (5.5%) patients reported a technical issue when connecting to telehealth.	Telehealth consultations were associated with out-of-pocket cost savings for 60% of respondents and included savings due to less time off work for themselves (19%) or their carer (1%), transport (49%), accommodation (7%), childcare (1%) and other (2%) costs.	Not recorded
Brenden Drerup et al. [12]	86 % of patients claimed that providers listened to concerns.	90.8 % of patients recommend this practice to others.	The most frequently suggested improvement from patients was related to improving technology, often quoted as the need for better internet connection and video quality (20% of patients)	Not recorded	the no-show (missed appointment) rate was 36.1% (56/155) compared with the telehealth visit no-show rate of 7.5% (14/186)
Fiona Imlach et al. [13]	98 % of surveyed patients claimed the physicians explain their concerns in easily way.	80% of patients would like to have telephone consultation and 69% would use video consultation in the future,	Not recorded	Not recorded	Not recorded
Ashwin Ramaswamy et al. [58]	patient satisfaction with video was significantly higher than in-person visits (94.9% vs. 92.5%, *p* < 0.001)	Not recorded	Not recorded	Not recorded	Not recorded
Tanya Ngo [57]	83 participants (96.5%) reported satisfaction with their care in terms of the amount of time the attending provider spent with them	75 patients (87.2%) stated that they would like to continue with telemedicine visits in the future.	Not recorded	76% of patients reporting that remote clinic visits made it possible for them to attend an appointment they would not otherwise have been able to.	Not recorded
Hyung Youl Park et al. [56]	87.1% of patients thought telemedicine had the same reliability as in-person visits.	85.1% of patients were willing to use telemedicine service again.	Not recorded	Almost 80% of patients reported the convenience of telemedicine	Not recorded
Lucinda Adams et al. [26]	The most of patients (75%) reported that their physician easily resolved their problems.	48.3% of respondents would continue to use telemedicine after COVID-19	50 % of patients rated telemedicine as “easy to use”.	75% of patients confirmed the convenience and time-saving due to telemedicine.	Not recorded
Mahta Mortezavi et al. [27]	74% of patients were satisfied with their telemedicine encounter.	Not recorded	3.6% of patients would have preferred an in-person visit instead because of technical issue.	Not recorded	Not recorded
Jesus Tornero – Molina et al. [28]	Patients were more satisfied with the telemedicine when their level of education were higher (OR = 4.12)	Not recorded	Not recorded	Not recorded	Not recorded
Matthew T. Jones et al. [29]	(84%) agreed their rheumatological health issues were satisfactorily addressed	169 of the questionnaire responders (60%) indicated they would be happy to have telephone consultations routinely in their future care.	Not recorded	Not recorded	Not recorded
Eleanor Layfield et al. [30]	Many of the patients who were willing to have another telemedicine visit agreed they felt telehealth met their needs.	Many participants (*n* = 25, 44.6%) also offered thoughts on the future use of telemedicine	Most patients found the connection process easy, whereas others reported technical challenges, including issues with connectivity and audio.	A common theme expressed by patients was that these visits were much more time efficient.	Not recorded
Janet S. Choi et al. [33]	Mean Press Ganey patient satisfaction scores during COVID-19 remained high at 94.5 (SD, 8.8; range, 20–100) for telemedicine visits	Not recorded	Not recorded	Not recorded	Not recorded
Phoebe Elizabeth Riley et al. [34]	The majority (78.8%) of patients felt that their provider had all the information needed to make a diagnosis and treatment.	Not recorded	Not recorded	Patients reporting a distance from their provider of 21 to 50 miles demonstrated an association with decreased overall satisfaction (odds ratio, 0.44; 95% CI, 0.24–0.82; *p* = 0.01) when compared with patients 0 to 20 miles and ≥50 miles from their provider.	Not recorded
M. Fieux et al. [35]	98% patients claimed that the physician had answered all of their questions.	The majority of patients (68%) were willing to use teleconsultation in the future.	Sound quality was judged poorly or not satisfactory by 24% of patients and video quality by 39%.	72% appreciated the time and cost savings.	Not recorded
Andrew M. Rizzi et al. [16]	Patients reported that surgeons demonstrated appropriate response to their concerns rating ‘good’ or ‘very good’ 95% of the time.	93% of patients reported they would participate in a telemedicine encounter again	Not recorded	Not recorded	Not recorded
Sandeep Kumar et al. [17]	92% of patients are satisfied with medical care they received during telemedicine	92% of respondents would recommend telemedicine to others	7.3% of patients found difficulty in understanding the process of teleconsultation	22.3 % found telemedicine as convenient	Not recorded
Lesslie J. Bisson et al. [18]	There were no difference in patient satisfaction between telemedicine and in-person orthopedic encounters during the COVID-19 pandemic.	Not recorded	Not recorded	Not recorded	Not recorded
Megan V. Morisada et al. [31]	There were no statistically significant differences in mean scores between the virtual visits and in-person visits with the interpersonal manner of the physician (*p* = 0.41), satisfaction with communication (*p* = 0.31), satisfaction with the time spent with doctor (*p* = 0.88).	Not recorded	There were no statistically significant differences in mean satisfaction scores between virtual visits and in-person visits with technical quality of care (*p* = 0.89)	There were no statistically significant differences in mean satisfaction scores between virtual visits and in-person visits with financial aspects of care (*p* = 0.89)	Not recorded
Firas Hentati et al. [32]	90% of patients claimed that thier needs were met during the telemedicine visit.	61.8% of patients were more likely to prefer the telehealth experience to being seen in-person.	The most commonly cited disadvantage to virtual visits was technological difficulties (17.8%).	Not recorded	Not recorded
S.M. Shahid et al. [52]	Forty-eight patients (91%) felt that they received the appropriate advice.	19 patients (36%) felt that telemedicine is not an appropriate platform to replace face-to-face consultations, after the COVID-19 pandemic	The satisfaction score for the clarity of the phone call was 4.9 out of 5.	Not recorded	Not recorded
Vidushi Golash et al. [53]	100% of respondents felt they were listened to by their doctor and had enough time to discuss their individual situation.	82.5% of patients who had a video consulation felt a face-to-face review would not have changed the outcome of their consultation.	57.5% of participants experienced no technical problems. The most common difficulty (20%) was problems with audio.	Not recorded	Not recorded
Alexander M. Satin et al. [44]	Overall, 87.7% of patients reported that they were satisfied with their telemedicine visit with 70% reporting a score of 5 out of 5 (“very satisfied”)	Not recorded	One-third of patients reported the issue with their telemedicine encounter-problems with video and audio (11.5%, and 9.7%, respectively).	Withmregard to mileage saved by a telemedicine visit, the majority of patients (56.9%) were within 25 miles round trip of their doctor’s office with a smaller subset of patients traveling over 100 miles (16.6%)	Not recorded
Sheena Bhuva et al. [45]	Overall, 83.7% of the patients were very satisfied with their telemedicine appointment and 13.9% were satisfied with the telemedicine appointment.	64.5% of the patients would rather have telemedicine over in-person appointments	Only 8% and 3% had issues with the video and audio, respectively	Not recorded	Not recorded
Karim Shafi et al. [46]	The majority of all patients noted that they were “extremely satisfied” with their treatment plan, with a mean score of 4.71 (SD = 0.55).	Not recorded	Ease of navigation was also scored highly, with a mean score of 4.32 (SD = 0.88). Of the 84 patients, only 2.4% noted that it was “very difficult” (1/5) to navigate the telehealth visit	Not recorded	Not recorded
Amol Raheja et al. [11]	Not recorded	33.3% of patients will prefer teleconsultation services over physical outpatient services even after its resumption.	Overall 15% (*n* = 35) patients faced difficulties during teleconsults.	Reduction of travel expenditure (*n* = 155, 67%), efficient utilization of time and resources for patients and their caregivers (*n* = 113, 49%) are the crucial advantages of telemedicine.	Not recorded
Ken Porche et al. [10]	Patients that underwent a telemedicine visit were more satisfied with the care provider’s explanations about their medical conditions, and were more satisfied with the concern the care provider showed for their questions compared to an in-person visit.	Not recorded	Not recorded	Not recorded	Not recorded
Alina Mohanty et al. [8]	91% of patients agreed that their provider satisfactorily addressed their clinical needs.	36% of patients stated they would like their future visits to be telehealth visits, with 48% patients stating they felt neutrally about this statement.	Not recorded	Not recorded	Not recorded
Elise J. Yoon et al. [9]	Patients rated the physician’s involvement in providing medical care as 6.70 ± 0.85.	The mean score for willingness to use telemedicine again was 5.56 ± 1.93 (mean ± SD)	The mean score for telemedicine visit equipment was 6.12 ± 1.55.	Not recorded	Not recorded
S. Shahzad Mustafa et al. [40]	77.4% patient believed telemedicine was as satisfactory as an in-person encounter.	Not recorded	4.2 % of patients experienced technical problems.	Not recorded	Not recorded
Kasey Lanier et al. [41]	93% of patients agree/strongly agree that the doctor explained their medical condition in a way they could understand.	46% preferred the teleconsultation over in-person visit in the future.	79% of patients claimed that connecting and starting telemedicine appointment was easy.	Not recorded	Not recorded
Judy Hamad et al. [14]	Almost 90% of patients rated the skillfulness of their providers as “very good” or “excellent”.	Logistic regression showed that prior telehealth experience was associated with higher odds of being willing to use teledermatology in the future (odds ratio [OR] 2.39, 95% CI 1.31–4.35; *p* = 0.004).	Patients’ satisfaction with visual quality was slightly higher than their satisfaction with voice quality and similar between follow-up patients and new patients.	Not recorded	Not recorded
Sumithra Jeganathan et al. [15]	Almost 60% of patients “strongly agree” and 30% of patient “agree” that the most of their questions and concerns were addressed.	78.3% would recommend telehealth visits to others	84.7% of patients found the process of connecting to their appointment easy	Not recorded	Telehealth visits also had a lower no-show rate; however, this difference was not statistically significant.
S. Glass Clark et al. [36]	The majority of patients answered either “agree” or “strongly agree” with the statement “All of my questions and concerns were addressed to my satisfaction during my video visit” (*n* = 89, 94.7%)	majority preferred to see the specialist in person, despite travel inconveniences, at their next visit (*n* = 66, 70.2%}	Not recorded	Not recorded	Not recorded
Kadri Haxhihamza et al. [38]	71 % of respondents rated their receiving medical care just about perfect.	Not recorded	Technical quality were rated the lowest of all aspect of telemedicine.	Not recorded	Not recorded
Devinder Kaur et al. [42]	98% of patients received adequate attention during the telephone follow-up session.	92% of patients would use telemedicine in the future, If the situation arose	98% of patients heard their physician clearly over the telephone.	All patients agreed telephone follow-up provides a timely and convenient service.	Not recorded
Zia Rahman et al. [43]	77.5% of patients remarked that their expectations were met	Not recorded	Not recorded	(69.4%) of the survey patients commented that the telephone consultation saved them time and money.	Not recorded
Sharon Orrange et al. [39]	90.1% (327/363) strongly agreed or agreed that the amount of time spent with the provider was adequate.	77.3% (279/361) reported that they “look forward to using telehealth in the future.”	Almost one-third of patients (114/365, 31.3%) had technical issues during the visit.	Not recorded	Not recorded
Allanah Smrke et al. [24]	Almost half (*n* = 42; 48%) would not want to hear bad news over the phone, with no difference based on age, sex, or education level.	The majority of patients (*n* = 86; 80%) indicated that they would like at least some future appointments to be performed using telemedicine	Not recorded	Common reasons for telemedicine preference were reduced travel time (*n* = 45; 42%), reduced travel expenses (*n* = 21; 20%) and convenience (*n* = 32; 30%).	Not recorded
Aniruddha Singh et al. [37]	Not recorded	120 respondents, 100 (83.0%) indicated they would at least consider using telehealth in the future, including 59 (49.2%) who said they were likely to or would prefer to use telehealth going forward.	Poor internet connectivity was of most concern, rated as at least somewhat of a factor by 35 (33.0%) respondents.	Reduced travel time was seen as a big advantage over traditional in-person appointments by 61 (57.5%) of the 106 respondents who participated in telehealth,	the no-show rate for telehealth visits (345/2019, 17%) was nearly identical to the typical no-show rate for in-person appointments
Agathoklis Efthymiadis et al. [55]	96% of patients were satisfied with their telemedicine appointment.	92 (77%) patients reported that they would like to use telephone consultation after the COVID-19 outbreak.	99% of patient rated the voice quality of appointment as “excellent” or “good”	98% of patients reported that the telephone consultation saved them time from travelling to the hospital.	Not recorded

## Data Availability

Not applicable.

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
