# Peer review of "Patient Satisfaction with Telemedicine during the COVID-19 Pandemic—A Systematic Review"

_ijerph, 2022, doi:10.3390/ijerph19106113_

Round 1

Reviewer 1 Report

The manuscript is overwhelmed by inherent presentations of study results regarding all the items starting with, Patient Satisfaction (misspelled satisfaction in the text) ending with, Willingness to use telehealth in the future.

I would suggest a table consisting of a ranking for each item depending on type of diseases. E.G. Prostate Cancer Clinic, the median degree of satisfaction was 9 on a 10-point scale, and The median patient satisfaction score of 5.5 (interquartile range [IQR] 4.25–6.25) was reported in oncology.

I would appreciate a table included in the manuscript with summary data (suggested above) that would increase the article comprehension. Please check English.

Author Response

Dear Reviewer,

We appreciate the comments as they pointed out how to improve the manuscript. 

Two tables summarizing our systematic review were prepared to increase comprehension of the article. Table number 1 summarizes all studies and the outcome i.e. patient satisfaction and table number 2 focuses on other aspects of patients’ evaluation of teleconsultation. Due to the comparatively length of the tables, we think the suitable place for them will be in supplementary materials.

Accordingly, we have reduced the text in the Results section. 

The manuscript has been checked by a proofreading professional.

The improved changes are marked by red.

We trust that the above changes will improve comprehension of the article.

We hope you will be satisfied with the response.

Thank you.

Reviewer 2 Report

Dear Authors,

your study is applied to an interesting topic, namely the telemedicine. This knowledge is crucial for forming appropriate future health policies and raises several interesting research questions on its economic impacts for the human health as well as on the health expenditure. Your review is applied to papers published during the COVID pandemic. However, you do not separate the papers referring to the pandemic and the pre-pandemic period. From the methodological point of view these two groups should be divided as it is important to find out whether results differ. Consequently, in your conclusions you claim “Telemedicine is undoubtedly a convenient device that can be used in the post-Covid-19 era” but on which findings you come to this conclusion? The pandemic era was a special period when not only the conditions to provide health services have changes but also the needs and expectations of the patients were different. Therefore, I think the paper could give important empirical evidence which would be able to give grounds to the policymakers. However, I do suggest important corrections regarding the methods used for forming the conclusions before accepting for publication.

Thank you for your research insights.

Author Response

Dear Reviewer, 

Thank you for your time spent reading our paper and your suggestions. The analysis referring to the pre-pandemic period were removed from our review to avoid methodological errors. It should be stressed that only two papers compared parameters between the pre-Covid period and the Covid era, and they did not contribute to obtaining significant conclusions.

The conclusion about the utility of telemedicine in the post-Covid period was too far-reaching. The frequency and willingness to use telemedicine among patients during the lifted restrictions should be assessed. We modified the conclusion “Telemedicine is undoubtedly a convenient device that ensured continuity of medical care during the Covid-19 pandemic, thanks to its diversified potential. “

The improved changes are marked by red.

We hope you will be satisfied with the response. 

We truly hope the paper will be found suitable for publication.

Thank you.

Reviewer 3 Report

This study is a review-type paper that discusses the various statuses of telemedicine in the current situation of Covid-19. It is a very timely and useful paper that conducted various analyses on telemedicine by a wide survey focusing on various papers previously published.

In order to improve the completeness and quality of this paper, please revise the following.
- The overall content is described in lengthy writing, which makes it less readable. In particular, it is recommended to use tables and illustrations in Chapters 3 (Results) and 4 (Discussion).
- In particular, it is recommended that Chapter 4 summarizes the advantages and disadvantages of telemedicine.
-Please describe the issue of personal information protection or personal information infringement, one of the important issues of telemedicine, in Chapter 4.
-The title of Section 3.2.2 is duplicated. Change to a different title (for example, Satisfaction rate)
-3.2.5. Willingness tu use telehealth in the future => Willingness to use telehealth in the future

Author Response

Dear Reviewer,

Thank you for your time spent reading our paper and your suggestions.

We appreciate the comments received greatly, as they pointed out how to improve the manuscript. Two tables summarizing our systematic review were prepared to increase comprehension of the article. The table number 1 summarizes all studies and the outcome i.e. patient satisfaction and table number 2 focuses on other aspects of patients’ evaluation of teleconsultation. Due to the comparatively length of the tables, we think the suitable place for them will be in supplementary materials.

Accordingly, we have reduced the text in the Results section. 

Information on data protection and patient privacy have been included in the discussion: 

“Security and protection of medical data, and patient privacy are also among the main concerns related to telemedicine [59]. There is no doubt that a secure computer system allows access to medical data to authorized medical professionals but prevents access to those who do not have the authorization. Another aspect that needs to be carefully assessed is obtaining verbally informed consent, which should be noted in the patient's medical records. [60] Effective verification and confirmation of patient identity is an essential part of ensuring patient privacy and preventing imposture. Due to the occurrence of cyberattacks in the past, physicians should inform patients about the risks associated with telehealth services. Telehealth platforms have healthcare-specific features and security, however, sometimes it is necessary to use instant video communication programs which are considered strictly secure by the U.S Department of Health and Human Services (HHS), e.g. Skype for Business, Microsoft Teams, VSee, Doxy.me. [61] For example, in the study by Mojdehbakh, surveyed patients claimed that their privacy was respected as either excellent (84.0%, n = 95) or good (8.8%, n = 10) during telemedicine. [21]”

The discussion was structured by highlighting the advantages and disadvantages of telemedicine.

The duplicate section title has been changed.

The manuscript has been checked by a proofreading professional.

The improved changes are marked by red.

We trust that the above changes will improve comprehension of the article.

We hope you will be satisfied with the response.

Thank you.

Round 2

Reviewer 1 Report

I think that the paper is overall better. Still, I suggest that table 1 should be organized in some manner e.g. studies per specialty.

Neurosurgery Clinic

Primary and Specialty Care Clinic

,

Author Response

I want to express my gratitude for your suggestions.
Table number 1 has been organized according to the specialty.

Reviewer 2 Report

The paper has significantly improved. 

Author Response

I want to express my gratitude for your comments on our systematic review.